

# Outstanding performance of an invasive alien tree *Bischofia javanica* relative to native tree species and implications for management of insular primary forests

Tetsuto Abe[1], Nobuyuki Tanaka[2] and Yoshikazu Shimizu[3]

[1] Kyushu Research Center, Forestry and Forest Products Research Institute, Kumamoto, Japan
[2] Department of Agricultural Science, Tokyo University of Agriculture, Tokyo, Japan
[3] Faculty of Arts and Sciences, Komazawa University, Tokyo, Japan

## ABSTRACT

Invasive alien tree species can exert severe impacts, especially in insular biodiversity hotspots, but have been inadequately studied. Knowledge of the life history and population trends of an invasive alien tree species is essential for appropriate ecosystem management. The invasive tree *Bischofia javanica* has overwhelmed native trees on Haha-jima Island in the Ogasawara Islands, Japan. We explored forest community dynamics 2 years after a typhoon damaged the Sekimon primary forests on Haha-jima Island, and predicted the rate of population increase of *B. javanica* using a logistic model from forest dynamics data for 19 years. During the 2 years after the typhoon, only *B. javanica* increased in population size, whereas populations of native tree species decreased. Stem diameter growth of *B. javanica* was more rapid than that of other tree species, including native pioneer trees. Among the understory stems below canopy trees of other species, *B. javanica* grew most rapidly and *B. javanica* canopy trees decreased growth of the dominant native *Ardisia sieboldii*. These competitive advantages were indicated to be the main mechanism by which *B. javanica* replaces native trees. The logistic model predicted that *B. javanica* would reach 30% of the total basal area between 2017 (in the eastern plot adjacent to a former *B. javanica* plantation) and 2057 (in the western plot distant from the plantation site), which is a maximum percentage allowing to eradicate under the present guideline of the National Forest. The results suggest immediate removal of *B. javanica* is required to preserve native biodiversity in these forests.

# INTRODUCTION

Invasive alien species have diverse impacts on biodiversity and ecosystems worldwide (*Chapin et al., 2000*; *Mack et al., 2000*; *Lockwood, Hoopes & Marchetti, 2007*; *Bellard, Cassey & Blackburn, 2016*). Invasive trees have a competitive advantage due to their fast growth rate (*Lamarque, Delzon & Lortie, 2011*) and act as ecosystem engineers by altering biological interactions, water runoff, litter quality, and nutrient cycling

Corresponding author
Tetsuto Abe, tetsuabe@ffpri.affrc.go.jp

(*Vitousek & Walker, 1989*; *Binggeli, 1996*; *Crooks, 2002*; *Lepš et al., 2002*; *Wiser et al., 2002*; *Meyer & Lavergne, 2004*; *Gaertner et al., 2014*; *Motard et al., 2015*). Such impacts on native ecosystems are amplified on oceanic islands owing to an inherent vulnerability to alien species (*D'Antonio & Dudley, 1995*; *Lonsdale, 1999*; *Sax, Gaines & Brown, 2002*; *Pyšek & Richardson, 2006*; *Kier et al., 2009*; *Walsh et al., 2012*). As examples of the serious consequences of an invasive tree species, *Miconia calvescens* attracts seed dispersers, shows high shade tolerance, and threatens native plant biodiversity in Pacific insular mesic forests (*Meyer & Florence, 1996*; *Medeiros et al., 1997*). The alien nitrogen fixer *Morella faya* changes nutrient cycling and alters development of the forest vegetation on Hawaiian volcanic lava flows (*Vitousek & Walker, 1989*). On Réunion Island, *Casuarina equisetifolia* disturbs primary succession on lava flows (*Potgieter et al., 2014*) and *Ligustrum robustum* subsp. *walkeri* can become the dominant woody species in natural forests on this island (*Lavergne, Rameau & Figier, 1999*). Despite many examples of their ecological impacts, research on invasive trees has not progressed sufficiently (*Richardson et al., 2014*), probably because of the long lifespan of trees, which leads to a long time-lag between the initial invasion and expansion in distribution (*Webster, Nelson & Wangen, 2005*; *Wangen & Webster, 2006*).

The expansion mechanism of invasive tree species is a critical research focus. Although encroachment of primary forests by invasive tree species is not common, it can cause a vegetation shift initially in canopy gaps that result from wind storms (*Knapp & Canham, 2000*; *Bellingham, Tanner & Healey, 2005*; *Brown, Scatena & Gurevitch, 2006*). Even without gap formation, shade-tolerant alien trees sometimes spread under the closed canopy of a mature native forest (*Wangen & Webster, 2006*; *Martin, Canham & Kobe, 2010*). Invasion of insular native forests by such alien tree species will exacerbate ecological deterioration of native forests in addition to the fragmentation caused by human activity since the initial colonization of the island (*Mueller-Dombois, 2008*). Given that ecosystem degradation generally progresses as alien species invade, a conservation plan should take into account the invasion rate. However, few case studies have estimated the rate of invasion from stand dynamics data (*Webster, Nelson & Wangen, 2005*). Additional studies of invasive tree species are needed to understand details of the invasion dynamics and rate of invasion (*Martin, Camham & Marks, 2009*; *Richardson & Rejmánek, 2011*; *Richardson et al., 2014*).

The Ogasawara Islands host insular ecosystems with high endemic biodiversity, but several invasive tree species are causing drastic changes to the vegetation (*Hata et al., 2006*; *Fukasawa et al., 2009*; *Abe, Yasui & Makino, 2011*). *Bischofia javanica* (Phyllanthaceae) is naturally distributed from Taiwan to Southeast Asia in the nearby area (e.g., *Lin et al., 2017*) but is an invasive alien tree species in the Ogasawara islands (*Yamashita et al., 2000*; *Shimizu, 2003*). The species is invasive in the mountainous area of the islands, which are covered by rich forest soils, with relatively high atmospheric humidity and frequent fogging (*Shimizu, 2003*; *Fukasawa et al., 2009*; *Tanaka et al., 2010*). *B. javanica* exhibits moderate shade tolerance, and can quickly shift photosynthetic mode between shade and direct sunlight (*Yamashita et al., 2000*). Such flexibility helps individuals to outcompete native trees after a disturbance event. The distribution of *B. javanica* on Haha-jima Island overlaps with that of mesic forests in which several endemic species are aggregated and thus poses

a serious threat to the native ecosystem. In contrast, the forests in the Sekimon area of Haha-jima have experienced minimal anthropogenic disturbance and thus still resemble the original primary mesic forests (*Shimizu, 2003*; *Abe, Tanaka & Shimizu, 2018*).

To develop effective eradication strategies for an invasive species for biodiversity conservation, its life history and population trend should be clarified (*Sakai et al., 2001*). We employed a permanent plot census, which is a standard method to describe forest dynamics (*Losos & Leigh, 2004*), and explored the dynamics of trees focusing on the relationships between alien and native species. Generally, the ecological risks posed by invasive tree species tend to be underestimated because of the usual lag period following their introduction (*Frappier et al., 2003*). Management of invasive alien species must be strategic to reduce the high social costs (*Higgins, Richardson & Cowling, 2000*; *Pimentel et al., 2000*). These observations suggest that appropriate prediction of the expansion of invasive tree species will contribute to effective forest management. In this study, we first investigated the short-term (2 years) dynamics to clarify the mechanism of aggressive invasion by *B. javanica* in insular primary forests on the Ogasawara Islands. As a result of an unexpected typhoon impact, the observed forest dynamics included responses to the disturbance and later crown shading. Second, we predicted the rate of expansion of *B. javanica* based on longer-term (19 years) population trends in the census plot. On the basis of our findings, we propose an effective strategy for forest management framed as a time limit for eradication.

## METHODS

### Study site

The oceanic Ogasawara Islands are located in a subtropical region of the Pacific Ocean (between 24°14′N and 27°44′N, and 140°52′E and 142°16′E). The resident biota contains a high percentage of endemic species (*Shimizu, 2003*). Haha-jima Island is one of the two inhabited islands in the archipelago. It covers 20 km$^2$ and has a maximum elevation of 463 m above sea level. The island's central mountains are covered by mesic forests that consist of relatively tall trees (about 15 m in height) compared with that of other forests in the Ogasawara Islands. The Sekimon mesic forests cover uplifted limestone in the northeastern corner of Haha-jima. The uplifted limestone has a doline-like central depression. Relatively thick sedimentary soil (*Okamoto et al., 1995*) and protection from wind by the walls of the doline have favored the growth of dense, tall forest on the base of the doline. This environment provides habitat for many plant species that the distributions are restricted to the Sekimon (*Abe, Tanaka & Shimizu, 2018*). *B. javanica* was introduced to the Ogasawara Islands for the silvicultural purpose before 1905 (*Toyoshima, 1938*; *Shimizu, 2003*). Although there is no record of planting *B. javanica* in the Sekimon in the forest management ledger, a participator attested that *B. javanica* had been planted before 1935 (*Toyoda, 2003*). In 1997, the seaward edge of the doline collapsed (Fig. 1A) and, subsequently, many trees have been exposed to salt-bearing onshore wind, causing desiccation and salt damage to the trees.

This area was struck by a strong typhoon in late 2006. Typhoon 0614 YAGI was spawned on 19 September in the northwestern Pacific (20.3°N, 159.2°E), about 1,800 km southeast

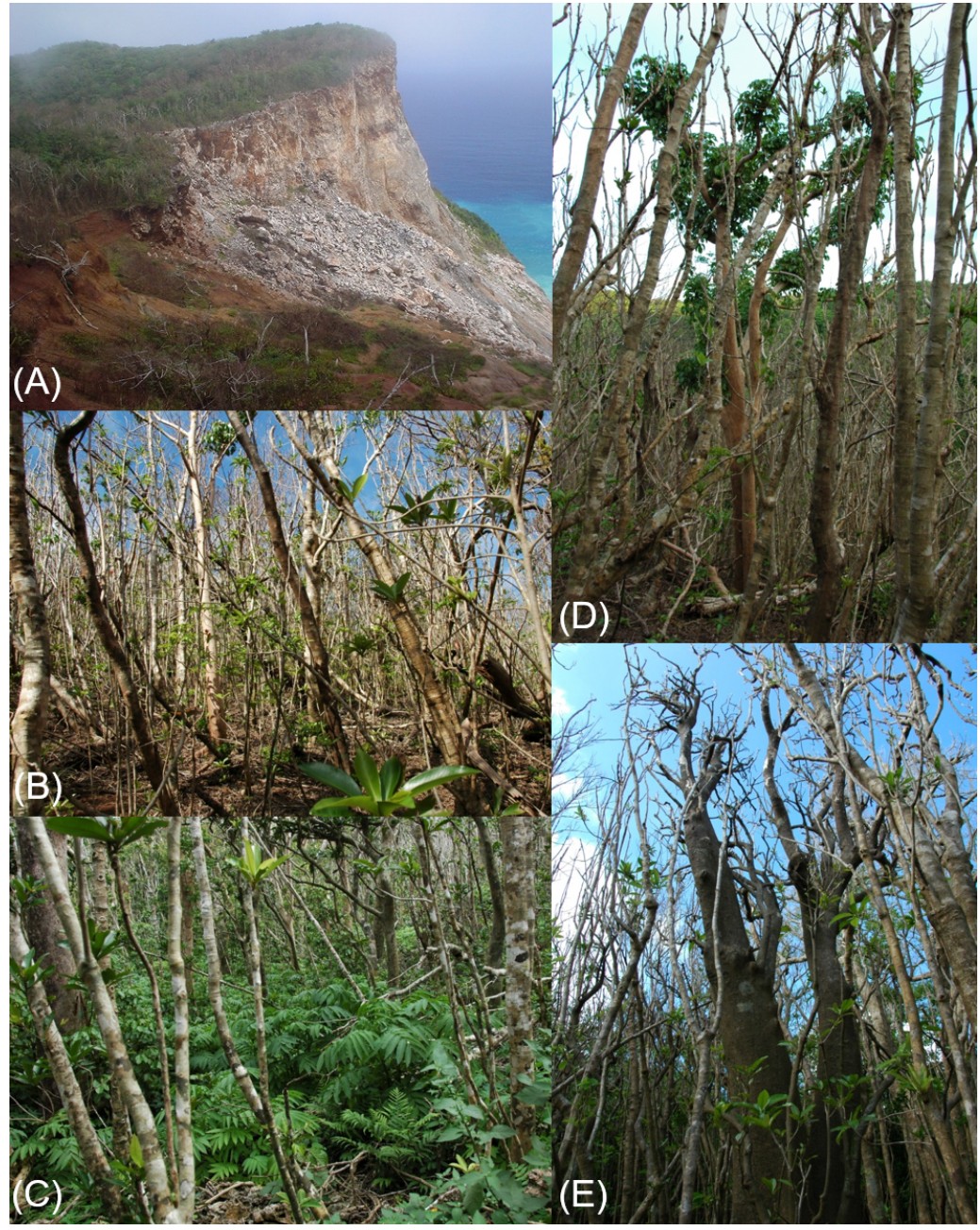

**Figure 1  Photographs showing the situation of the Sekimon forests after the typhoon.** (A) View of the mesic forests on the Sekimon uplifted limestone on 4 October 2006. The southern part of the uplift collapsed in a landslide in 1997. (B) Defoliation of canopy crowns by typhoon 0614 YAGI (22 November 2006). (C) Regeneration of *Sambucus chinensis* var. *formosana* on the sunny forest floor after the typhoon (17 April 2007). (D) Rapid flushing of *Bischofia javanica* after the typhoon damage (22 November 2006). (E) Defoliated crowns of *Pisonia umbellifera* and *Ardisia sieboldii* (22 November 2006).

of the Ogasawara Islands. The typhoon was closest to Haha-jima Island on 22 and 23 September when it passed about 100 km west of Haha-jima. At that time, the atmospheric pressure decreased to 930 hPa, the maximum wind velocity attained 45 m s$^{-1}$, and the 170 km radius of storm area experienced a wind velocity $\geq$ 25 m s$^{-1}$ estimated by the Dvorak method (*Japan Meteorological Agency, 2018*).

**Field survey**

We selected a survey area in the central portion of the primary forests in the Sekimon area and established two 2-ha census plots (100 m $\times$ 200 m) because there is a steep limestone ridge difficult to traverse between the two plots. We surveyed all trees with diameter at breast height (DBH) $\geq$ 10 cm in 2006 and described the status of each individual's crown in terms of whether it formed part of the forest canopy or understory. We defined canopy trees as individuals in which more than half of the crown surface was exposed to direct sunlight (i.e., not shaded by neighboring trees); for individuals classified as an understory tree, we recorded the tree species that covered the largest proportion of its crown. This judgement was conducted by eyesight, aided by observation using binoculars when necessary. In 2008, we conducted a second census following the same method of the first census. The abbreviations shown in Table 1 were used for the species names used in the figures and tables in this paper.

*Shimizu (1994)* surveyed a portion of our study site in 1987 using a 100 m $\times$ 50 m plot (Fig. S1). The southern portion of this plot disappeared in a landslide in 1997 (Fig. 1A). The present study plot included the remaining portion (60 m $\times$ 50 m) of the Shimizu plot in the southeastern part of the western plot. Our reconstruction of the Shimizu plot was based on a tree-by-tree map drawn in 1987 (*Shimizu, 1994*). We checked the position of characteristic large trees (e.g., *Melia azedarach*) and old stumps of *Morus boninensis* that had been cut about 130 years previously but had not decomposed because of the strong, decay-resistant wood (*Yoshida & Oka, 2000*). The 1987 data enabled us to analyze changes in species composition in terms of the number of stems and basal area. However, we could not analyze individual mortality and growth since 1987 because *Shimizu (1994)* did not label individual trees.

To detect the impacts of typhoon 0614 YAGI, we surveyed the damage soon after the first tree census (November and December 2006). We recorded the types of damage for individual trees with DBH $\geq$ 10 cm in the northern half of the western plot (1 ha, $N = 2675$). The damage to each tree was classified as defoliated, snapped, uprooted, or trapped (under one or more uprooted trees). Among these damaged trees, the stems that died at the 2008 survey were judged to have died due to typhoon damage, and the mortality rate was defined as the number of the dead stems in 2008 divided by the number of stems in 2006 damage survey.

Field survey was approved for the Ogasawara National Park by the Ministry of Environment (No. 0606328007, No.080507006) and for the Ogasawara National Forest by the Forest Agency (No.18-2-50 and No.20-1-32).

**Table 1** **Abbreviations for tree species names.** Species order is based on APG III (*Yonekura & Murata, 2012*).

| Family | Species | Species abbr. |
|---|---|---|
| Cyatheaceae | *Cyathea mertensiana* | Cyme |
| | *C. spinulosa* | Cysp |
| Lauraceae | *Cinnamomum pseudopedunculatum* | Cips |
| | *Machilus boninensis* | Mabo |
| | *M. kobu* | Mako |
| | *Neolitsea sericea* var. *aurata* | Nese |
| | *N. boninensis* | Nebo |
| Pandanaceae | *Pandanus boninensis* | Pabo |
| Arecaceae | *Livistona boninensis* | Libo |
| Rosaceae | *Rhaphiolepis indica* var. *umbellata* | Rhin |
| Cannabaceae | *Celtis boninensis* | Cebo |
| | *Trema orientalis* | Tror |
| Moraceae | *Ficus boninsimae* | Fibo |
| | *F. iidana* | Fiii |
| | *Morus australis* | Moau |
| | *M. boninensis* | Mobo |
| Elaeocarpaceae | *Elaeocarpus photiniifolius* | Elph |
| Euphorbiaceae | *Claoxylon centinarium* | Clce |
| Phyllanthaceae | *Bischofia javanica* | Bija |
| Putranjivaceae | *Drypetes integerrima* | Drin |
| Myrtaceae | *Syzygium cleyerifolium* | Sycl |
| Rutaceae | *Melicope grisea* var. *grisea* | Megr |
| | *Zanthoxylum ailanthoides* var. *inerme* | Zaai |
| Meliaceae | *Melia azedarach* | Meaz |
| Malvaceae | *Hibiscus glaber* | Higl |
| Caricaceae | *Carica papaya* | Capa |
| Nyctaginaceae | *Pisonia umbellifera* | Pium |
| Sapotaceae | *Planchonella obovata* var. *obovata* | Plob |
| Primulaceae | *Ardisia sieboldii* | Arsi |
| Rubiaceae | *Gardenia boninensis* | Grbo |
| | *Psychotria homalosperma* | Psho |
| Loganiaceae | *Geniostoma glabrum* | Gegl |
| Apocynaceae | *Ochrosia nakaiana* | Ocna |
| Oleaceae | *Ligustrum micranthum* | Limi |
| Lamiaceae | *Callicarpa subpubescens* | Casu |
| Aquifoliaceae | *Ilex mertensii* var. *beechyi* | Ilmb |
| | *I. mertensii* var. *mertensii* | Ilmm |

## Statistical analyses

We evaluated the annual diameter growth rate in 2-year period as ((DBH in 2008)-(DBH in 2006))/(survey interval months)*12/(DBH in 2006)*100 for each tree species. The morality rate of each tree species was defined as the number of dead stems in the 2008

survey divided by the number of stems in 2006 survey. The population growth rate was defined as the period growth rate of the number of stems: (N in 2008)-(N in 2006)/(N in 2006)*100, where N is the number of stems. Generally, trees have a trade-off relationship between growth and survival (*Grubb, 1977*; *Hubbell & Foster, 1992*; *Wright et al., 2003*), but *B. javanica* on Hahajima Island seemed to have good performance for both. To confirm this, the Pearson's product-moment correlation coefficient between the annual diameter growth rate and the population growth rate was examined when all tree species were used and when only *B. javanica* was removed.

Differences between *B. javanica* and native trees for typhoon damages and stem dynamics (mortality and recruitment) were examined by a Tukey's HSD multiple comparison after generalized linear model (GLM) analyses using the multcomp package in R ver. 3.3.2 (*R Core Team, 2016*). The GLMs of typhoon damage were conducted independently for each type of damage and mortality assuming a binomial error distribution with the number of damaged stems as a responsible variable and the tree species as an explanatory variable. The GLMs of population growth were conducted assuming a binomial error distribution with the number of recruited stems or the number of dead stems as a responsible variable and the tree species as an explanatory variable, respectively. We examined the effects of crown position on diameter growth of understory tree stems using two types of analysis: the effect of the canopy tree species on a given understory species and the growth differences among the understory tree species under a given canopy species. Both analyses used a general linear model (GLM) with a Gaussian link function and a multiple-comparison test using R. The responsible variable was the annual diameter growth rate of understory tree stems in both GLM analyses. The explanatory variable was understory tree species in the comparison among understory species under a given canopy species and was canopy tree species in the comparison among canopy species over a given understory species.

In the tree invasion process, it is effective to cover the understory trees with a wide crown in addition to the fast growth. Even if individual understory stems are likely to die sooner or later, there are always many stems under the wide canopy in the process of development of canopy trees, and conversely there would be only fewer stems with more than 10 cm DBH under the narrow canopy. Since we did not directly measure individual crown widths, we used, simply assuming that there are many stems under the wide crown, the following formula to index the crown area (CW) of each tree species:

$$CW = NS/NC$$

where NS is the number of stems covered by the crown of the canopy species and NC is the number of canopy stems of the species.

## Prediction of increase in *B. javanica* occupancy

It is preferable to use highly accurate models, such as a population matrix, to predict the population dynamics of an invasive tree species (e.g., *Buckley, Briese & Rees, 2003*). However, we could not use such a model in the present analysis because we surveyed the young trees less than 10 cm in DBH including seedlings only once (*Abe, Tanaka & Shimizu, 2018*). Instead, we used a simple logistic curve (*Radosevich, Stubbs & Ghersa, 2003*; *Webster*
& *Wangen, 2009*) to predict future population growth of *B. javanica* in terms of the number of stems and basal area. Given that it can be assumed that the spread of an invasive tree species is random and continuous within the forest, a simple model prediction is considered to be sufficiently practicable (*Frappier et al., 2003*). The model represented the proportion of *B. javanica* ($D_{BJ}$) with an upper limit of 1.0 for the proportion, as follows:

$$D_{BJ} = 1/\{1 + a \times \exp(-b \times t)\}$$

where *t* represents the number of years since 2006. The coefficients *a* and *b* were determined based on the data from the 1987 measurements in the *Shimizu (1994)* plot and the 2006 measurements in *Abe, Tanaka & Shimizu (2018)* (Table S1). Although the two plots were separated for convenience because of the cliff between them, the vegetation of both plots is considered to be homogeneous. Accordingly, we applied these parameters to the prediction of *B. javanica* dynamics in both plots.

We predicted the time required for *B. javanica* to attain 30% and 50% of the number of stems and basal area for the western plot and eastern plot, using logistic regression models. The lower percentage (30%) was based on the guideline of the National Forest that restricts the proportion of tree removal less than 30% of the total volume to prevent soil erosion. The higher percentage (50%) was based on data from the forests on Mt Kuwanoki (Haha-jima Island), where the former forest type had been identical to that at the Sekimon but now resembles a *B. javanica* forest stand with more than 40% occupancy of the total basal area (*Shimizu, 1988*). In addition, as a property of the logistic model, the estimated year tends to include a smaller error in the central portion of the logistic curve (e.g., between 30% and 70% occupancy) than that at each extreme (i.e., the first year of invasion and the end of the simulation period). Therefore, forecast years reaching 30% and 50% occupancy are expected to be most accurate and robust.

## RESULTS

### Survival, growth, and typhoon damage

Typhoon 0614 YAGI was situated closest to Haha-jima Island on 22 and 23 September 2006. The typhoon defoliated all standing stems (Fig. 1B), and snapped, uprooted, and trapped trees accounted for 6.9%, 2.6%, and 0.2% of the total, respectively (Table 2). There was no significant difference in the proportion of stems of these types of typhoon damage between native species and *B. javanica*. Pioneer trees (sun-lit trees growing rapidly in the early stage of succession or in the gaps) exhibited relatively high mortality (*Zanthoxylum ailanthoides* var. *inerme* at 16.7%, *Trema orientalis* at 33.3%, and *Cyathea mertensiana* at 21.4%), as did some later-successional species (*Ochrosia nakaiana* at 50.0% and *Psychotria homalosperma* at 21.4%). *B. javanica* showed low mortality (1.9%) in response to the typhoon disturbance.

The number of stems decreased between 2006 and 2008 among the most frequent tree species (more than 30 stems in the plots) except for *B. javanica* (7.4% increase) (Fig. 2). The increment in *B. javanica* was the result of recruitment of 44 individuals to the DBH $\geq$ 10 cm size class and the death of 10 individuals. Species that showed the greatest decrease

**Table 2  Numbers of trees damaged by typhoon 0614 YAGI.** "Uprooted" includes inclined individuals with at least half of the root system exposed. Values of the number of damaged stems are "the number of damaged stems including dead stems"/"the number of dead stems" in 1 ha area.

| Species | Origin | N | The number of damaged stems | | | |
|---|---|---|---|---|---|---|
| | | | Defoliated | Snapped | Uprooted | Trapped |
| Cyme | E | 28 | 28/1 | 4/3 | 2/2 | 0/0 |
| Cysp | E | 8 | 8/1 | 0/0 | 0/0 | 0/0 |
| Mabo | E | 75 | 75/5 | 11/2 | 2/0 | 0/0 |
| Mako | E | 8 | 8/0 | 3/0 | 0/0 | 0/0 |
| Rhin | I | 2 | 2/0 | 0/0 | 0/0 | 0/0 |
| Cebo | E | 23 | 23/0 | 0/0 | 0/0 | 0/0 |
| Tror | I | 3 | 3/1 | 0/0 | 0/0 | 0/0 |
| Fibo | E | 51 | 51/3 | 1/0 | 4/3 | 0/0 |
| Moau | A | 2 | 2/0 | 0/0 | 0/0 | 0/0 |
| Elph | E | 208 | 208/12 | 20/5 | 7/1 | 1/0 |
| Bija | A | 54 | 54/1 | 3/0 | 4/0 | 0/0 |
| Sycl | E | 12 | 12/0 | 1/0 | 0/0 | 0/0 |
| Megr | E | 96 | 96/8 | 3/2 | 0/0 | 0/0 |
| Zaai | E | 6 | 6/1 | 0/0 | 0/0 | 0/0 |
| Meaz | I | 1 | 1/0 | 0/0 | 0/0 | 0/0 |
| Higl | E | 27 | 27/0 | 3/1 | 4/1 | 0/0 |
| Pium | I | 56 | 56/0 | 2/0 | 2/0 | 0/0 |
| Plob | I | 81 | 81/2 | 3/1 | 9/1 | 1/0 |
| Arsi | I | 1,985 | 1985/149 | 132/35 | 34/19 | 3/0 |
| Grbo | E | 1 | 1/0 | 0/0 | 0/0 | 0/0 |
| Psho | E | 28 | 28/4 | 3/1 | 2/1 | 0/0 |
| Ocna | E | 4 | 4/2 | 0/0 | 0/0 | 0/0 |
| Limi | E | 1 | 1/0 | 0/0 | 0/0 | 0/0 |
| Casu | E | 5 | 5/0 | 1/0 | 2/1 | 0/0 |
| Total | | 2,765 | 2765/190 | 190/50 | 72/29 | 5/0 |

**Notes.**

E, Endemic to the Ogasawara Islands; I, indigenous; A, alien for the Ogasawara Islands.

Typhoon YAGI was situated closest to Haha-jima Island on 22 September 2006 and the survey was conducted in November and December 2006. Abbreviations of species name are defined in Table 1.

in number of stems were an endemic pioneer, *Z. ailanthoides* var. *inerme* (−43.3%), and an endemic tree fern, *Cyathea mertensiana* (−34.8%). The proportion of the number of recruitments into the stem size class DBH ≥ 10 cm was largest for the alien species *B. javanica* (8.8%) followed by *Callicarpa subpubescens* (6.9%) and *Ficus boninsimae* (6.6%). Some native species had a significantly higher proportion of the number of dead stems and significantly less proportion of the number of recruitments than *B. javanica* (Fig. 2). Annual diameter growth rate (Fig. 3) was largest in *B. javanica* (3.1 ± 0.1%, mean ± SE) followed by three pioneers, *C. mertensiana* (2.1 ± 0.4%), *Z. ailanthoides* var. *inerme* (2.1 ± 0.3%), and *C. subpubescens* (2.0 ± 0.3%). The diameter growth rates of dominant native species were less than half that of *B. javanica* (e.g., *Ardisia sieboldii* at 0.8 ± 0.0%, *Elaeocarpus photiniifolius* at 1.0 ± 0.1%, and *Pisonia umbellifera* at 1.3 ± 0.1%). Annual

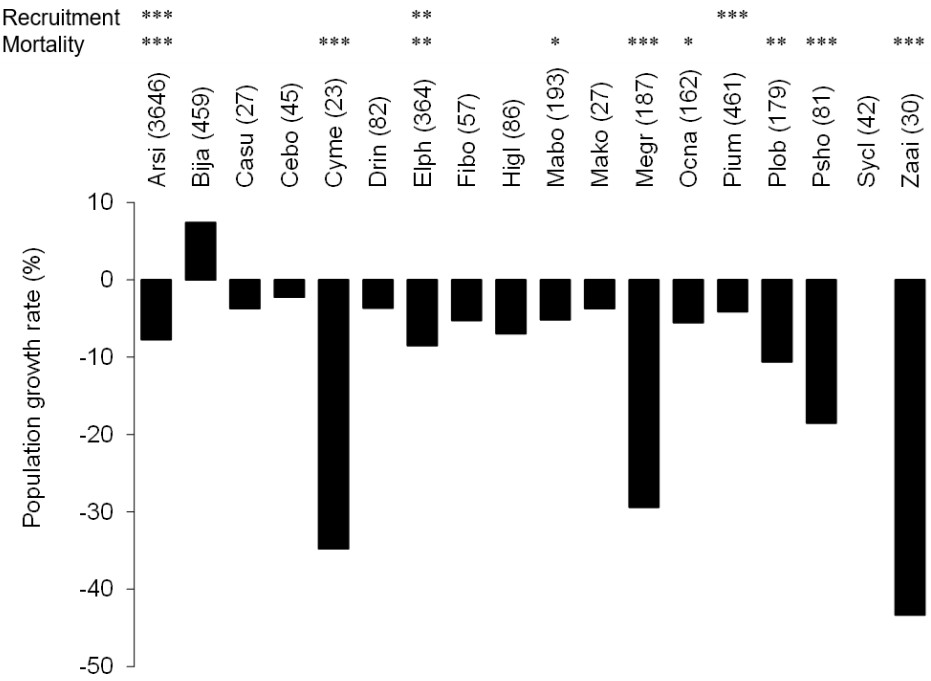

**Figure 2** **Population growth rates (individuals of DBH ≥ 10 cm) of the most frequent tree species between 2006 and 2008.** Values within parentheses after the species names represent the number of stems within the survey area (4 ha) in 2006. The significant differences of the proportion of dead and recruited stems between native species and *B. javanica* are shown at the top. In the tree species with significant difference, recruitments were all less than that of *B. javanica* and deaths were all more than that of *B. javanica*. ***, $p < 0.001$; **, $p < 0.01$, *, $p < 0.05$. Abbreviations for species names are defined in Table 1.

diameter growth rate was negatively correlated with population growth rate when the data for *B. javanica* were omitted from those for the most frequent tree species (Pearson's product-moment correlation, $r = -0.635$, $t = -3.182$, $df = 15$, $p = 0.006$), but no significant relationship was observed when the data for *B. javanica* were included ($r = -0.225$, $t = -0.922$, $df = 16$, $p = 0.370$).

## Effects of crown shading

The number of trees in which more than half of the crown was shaded by the crown of a neighboring tree in 2008 was 2761 (39.9% of all stems, Fig. 4); the number was largest for *A. sieboldii* (1956), *P. umbellifera* (301), and *B. javanica* (105). The most frequent canopy species were *E. photiniifolius* (793), *B. javanica* (685), and *Celtis boninensis* (219).

The mean annual diameter growth of understory trees was significantly less than that of canopy trees (GLM with a Gaussian link function; estimate = 0.059, $t = 8.32$, $P < 0.001$). The canopy of *B. javanica* significantly decreased the diameter growth of several understory tree species: diameter growth was significantly decreased for *A. sieboldii* than under *E. photiniifolius* and under *Z. ailanthoides* var. *inerme*, and for *P. umbellifera* than under *A. sieboldii* (Fig. 5). On the other hand, understory individuals of *B. javanica* exhibited superior growth compared with that of native understory tree species, regardless of the canopy tree species (Fig. 6). Although the CW index was much larger in *M. azedarach*
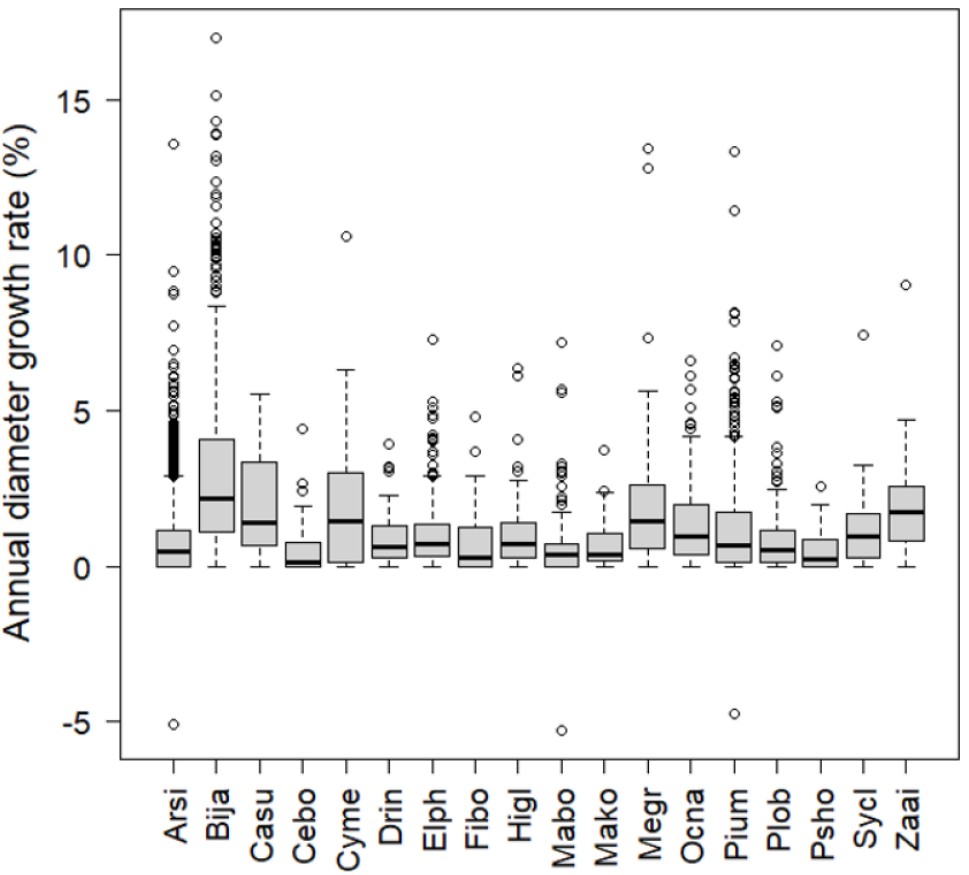

**Figure 3  Annual diameter growth rate from 2006 to 2008.** The thick line in the center of the boxplot shows the median value of the data. The top of the box represents the third quartile and the bottom of the box represents the first quartile. Circles represent outliers. Abbreviations of species name are defined in Table 1.

(CW = 5.3) and *C. boninensis* (4.9) compared with that of all other species (Fig. 7), the largest values of CW among dominant species (i.e., those with ≥ 100 canopy individuals) were for *E. photiniifolius* (2.2), followed by *B. javanica* (1.9) and *Planchonella obovata* var. *obovata* (1.1). The most frequent dominant species, *A. sieboldii*, showed a small CW index (<0.1).

## Prediction of invasion by *B. javanica*

In the Shimizu plot, *B. javanica* increased substantially in both the number of stems (176.4%) and basal area (177.8%) for the 19-year period (Table S1). We applied these changes for *B. javanica* to estimate the coefficients of logistic curves (Fig. 8). The coefficients of the logistic model were $a = 36.214$ and $b = 0.038$ based on the number of stems, and $a = 36.155$ and $b = 0.051$ based on the basal area. The model predicted that in the eastern plot, *B. javanica* will account for 30% of the number of stems in 2033 and 30% of the basal area in 2017. In the eastern plot, *B. javanica* will account for 30% of the number of stems in 2087 and 30% of the basal area in 2057. In the eastern plot, *B. javanica* will account for

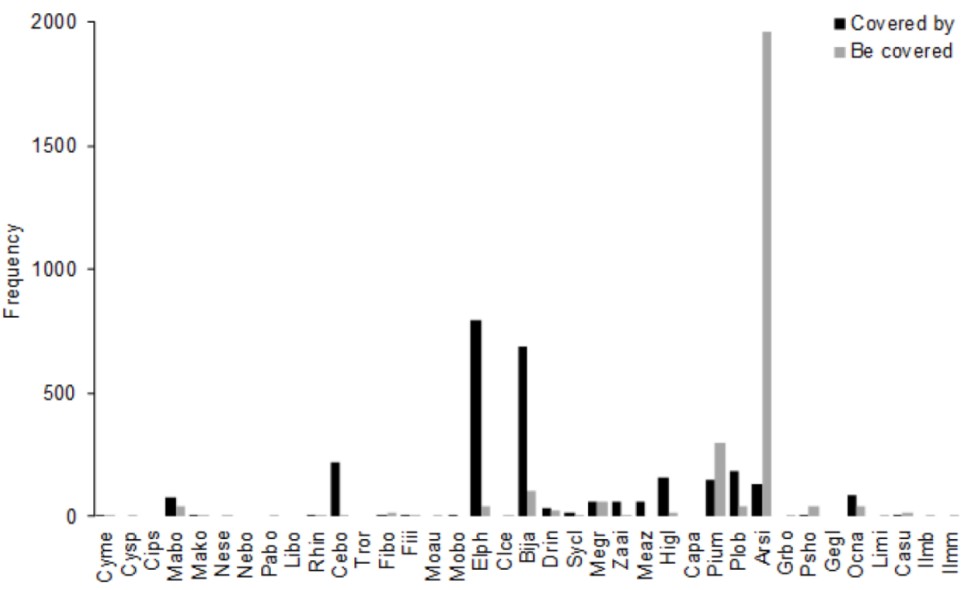

**Figure 4** **Frequency of crown positions in the 4 ha survey area in 2008.** "Covered by" is the total number of understory stems (DBH ≥ 10 cm) that the species covered. "Be covered" is the number of understory stems of the species that the crown is covered by other trees, including conspecifics. Abbreviations of species name are defined in Table 1.

50% of the number of stems in 2056 and 50% of the basal area in 2034. In the western plot, *B. javanica* will account for 50% of the number of stems in 2109 and 50% of the basal area in 2074.

## DISCUSSION

The invasive tree species *B. javanica* showed increased performance relative to native trees after typhoon 0614 YAGI. The diameter growth rate and survival rate of *B. javanica* were higher than those of other tree species in the study plots, including native pioneer trees. Given that rapid growth is a strong indicator of invasiveness (*Lamarque, Delzon & Lortie, 2011*), *B. javanica* showed high invasive ability in the Sekimon area of Haha-jima Island. In addition, *B. javanica* showed the most rapid leaf flush after defoliation by the typhoon (Fig. 1D). Since the size distribution of adult *B. javanica* trees was richest in the smallest size class and the seedlings in the forest floor was frequent (*Abe, Tanaka & Shimizu, 2018*), its recruitment is presumed to be high. As a result, *B. javanica* increased in population size after the typhoon, whereas native tree species decreased in population size. Dominant native tree species mostly ceased diameter growth for two years while pioneer trees showed larger diameter growth rate. The negative correlation between diameter growth rate and population growth rate among the dominant native tree species is likely to reflect the well-known growth–survivorship trade-off (*Grubb, 1977*; *Hubbell & Foster, 1992*; *Wright et al., 2003*). However, *B. javanica* showed exceptional positive population growth despite the rapid diameter growth. This difference may be the result of an inherent vulnerability to invasive species on oceanic islands that exhibit a high percentage endemicity

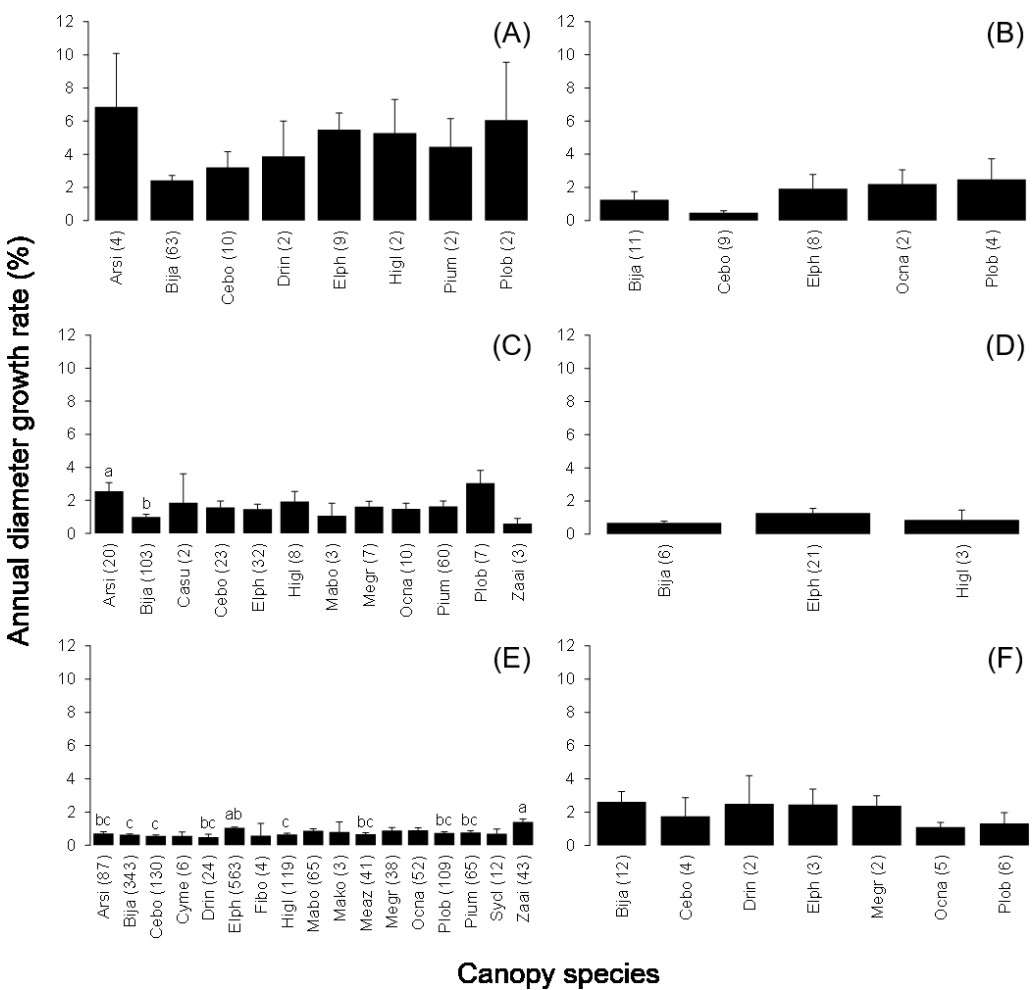

**Figure 5  Annual diameter growth rate in the six most frequent tree species under canopy trees.** Understory species are (A) Bija, (B) Plob, (C) Pium, (D) Elph, (E) Arsi and (F) Ocna. The stem diameter was measured at breast height. Values within parentheses represent the number of canopy individuals. Bars labeled with different letters differ significantly (*P* < 0.05, Tukey–Kramer test). Error bars represent the SE. Abbreviations of species name are defined in Table 1.

(*Berglund, Järemo & Bengtsson, 2009*; *Walsh et al., 2012*). Windstorm disturbance usually creates the opportunity for invasive plant species to spread in natural insular forests (*Fine, 2002*; *Denslow, 2003*; *Lugo, 2004*; *Bellingham, Tanner & Healey, 2005*).  A high number of seedlings of *B. javanica* and two additional alien species, *Carica papaya* and *Morus australis*, were observed on the Sekimon forest floor (*Abe, Tanaka & Shimizu, 2018*). This observation suggests that these alien species show high propagule pressure. In particular, seedlings of *B. javanica* show high photosynthetic plasticity (*Kamaluddin & Grace, 1992*; *Yamashita et al., 2000*), which can promote their acclimation to a range of light environments and permit a rapid growth response after forest disturbance (*Pattison, Goldstein & Ares, 1998*). Therefore, the seedlings of *B. javanica* are likely to exhibit greater percentage survival than native species after typhoon disturbance. Subsequently, young understory stems of *B.*

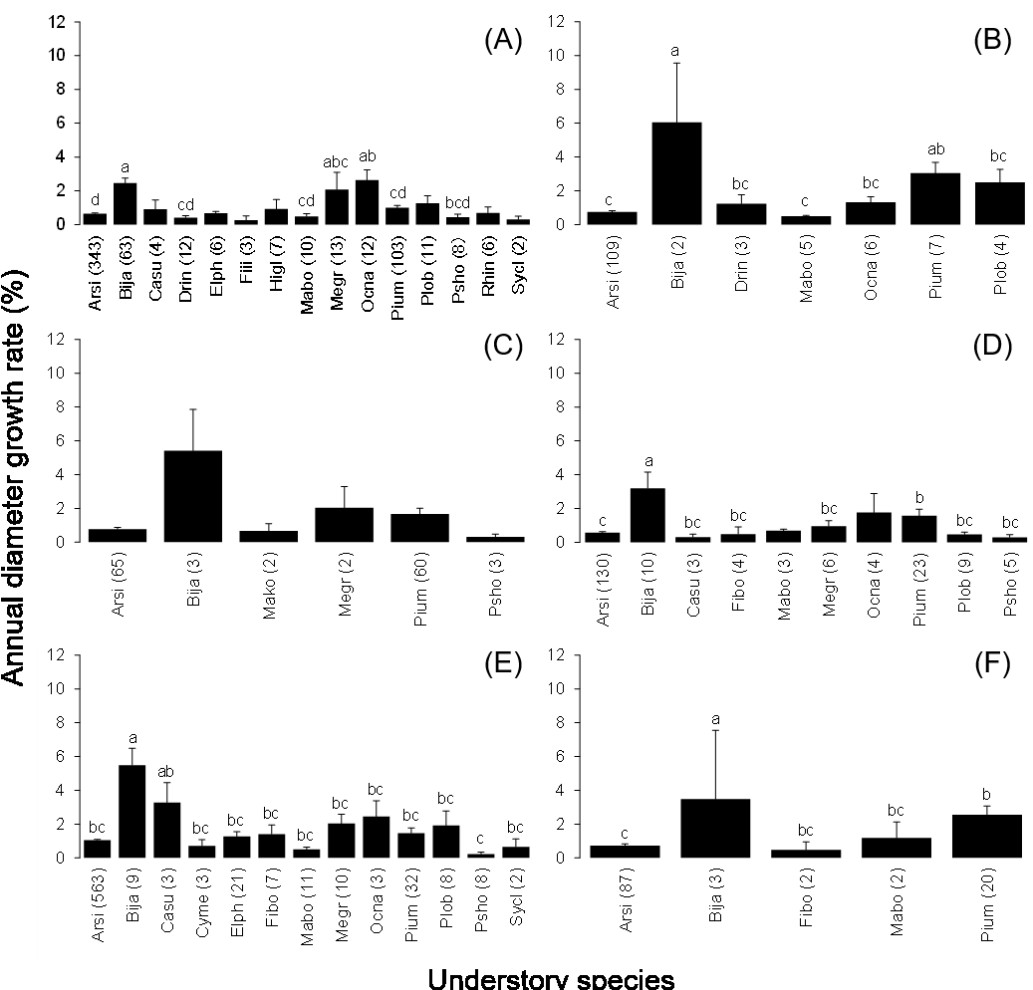

**Figure 6** **Annual diameter growth rate of stem diameter under the crown of the six most frequent tree species.** Canopy species are (A) Bija, (B) Plob, (C) Pium, (D) Cebo, (E) Elph and (F) Arsi. The stem diameter was measured at breast height. Values within parentheses represent the number of understory stems. Bars labeled with different letters differ significantly ($P < 0.05$, Tukey–Kramer test). Error bars represent the SE. Abbreviations of species name are defined in Table 1.

*javanica* grew more rapidly than understory individuals of native tree species regardless of the canopy tree species (Fig. 6).

The invasion rate of *B. javanica* was relatively slow in the Sekimon forests probably because the species is still in an early stage of invasion compared to other forests in the Ogasawara Islands. The number of stems and basal area of *B. javanica* increased by 1.4 times and 1.7 times, respectively, during the 19-year period in the Sekimon forests, whereas basal area of *B. javanica* increased to 9 times the 1984 value during the subsequent 19 years and overwhelmed the native tree species in secondary forests on Chichi-jima Island, located 50 km north of Haha-jima (*Hata et al., 2006*). Even in the early stage of invasion, the rate of increase of *B. javanica* in the Sekimon forests has exceeded those of native tree species, even though native species also have increased over the 19 years (Table S1). During

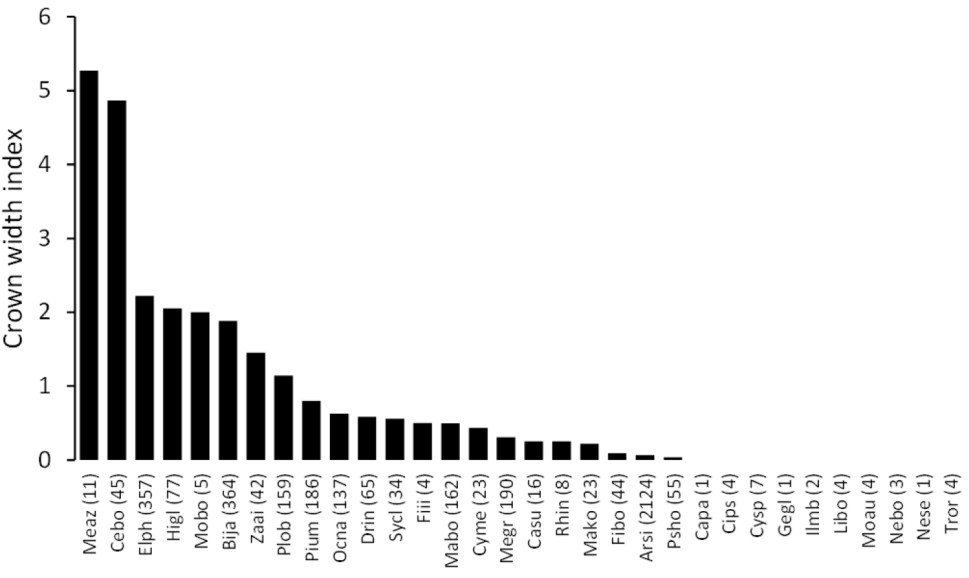

**Figure 7  Crown width index values for the tree species in the survey area.** Abbreviations of species name are defined in Table 1.

this period, typhoons with a wind speed of more than 20 m s$^{-1}$ struck 12 times and more than 30 m s$^{-1}$ struck four times in the Ogasawara Islands (Table S2). A preliminary study of the Sekimon forests also reported significant damage to the forest by a severe typhoon in 1983 (*Shimizu, 1994*). Repeated wind-induced disturbance is likely to have assisted the spread of *B. javanica* in the Sekimon forests.

Regarding crown position, the two dominant tree species, *A. sieboldii* and *P. umbellifera*, grew less under a *B. javanica* crown than those under *E. photiniifolius* and *A. sieboldii* crowns, respectively. Given that the defoliation damage caused by typhoon 0614 YAGI had recovered in 2008, the stem growth during the preceding two years included the effects of both typhoon disturbance and later crown shading, which are difficult to distinguish. A lower diameter growth rate under a *B. javanica* crown is partly due to the more rapid recovery of *B. javanica* crowns after the typhoon damage (Fig. 1D). In addition, *B. javanica* showed a relatively high CW, whereas few native tree species showed a high CW in the Sekimon forests. The dominant species *A. sieboldii* is a sub-canopy tree and develops a narrow crown. The tree species with a wide crown have a relatively deep crown (e.g., *Aiba & Kohyama, 1997*), and its understory would be poor light condition. Accordingly, although we did not measure the difference of light condition, it is assumed that *B. javanica*, which has a high CW suppress more understory stems than many native trees with low CW. This may be the reason why *P. umbellifera* individuals showed superior growth under *A. sieboldii* crowns than under *B. javanica* crowns. Other native tree species (e.g., *Machilus boninensis*, *Melicope grisea* var. *grisea*, *O. nakaiana*, and *P. umbellifera*) also produce narrow crowns and are likely to have similar effects on understory trees that we may have failed to detect (Fig. 5) because of the small sample sizes. Although spatiotemporal variation in forest structure caused by wind-induced disturbance is an important mechanism of tree species

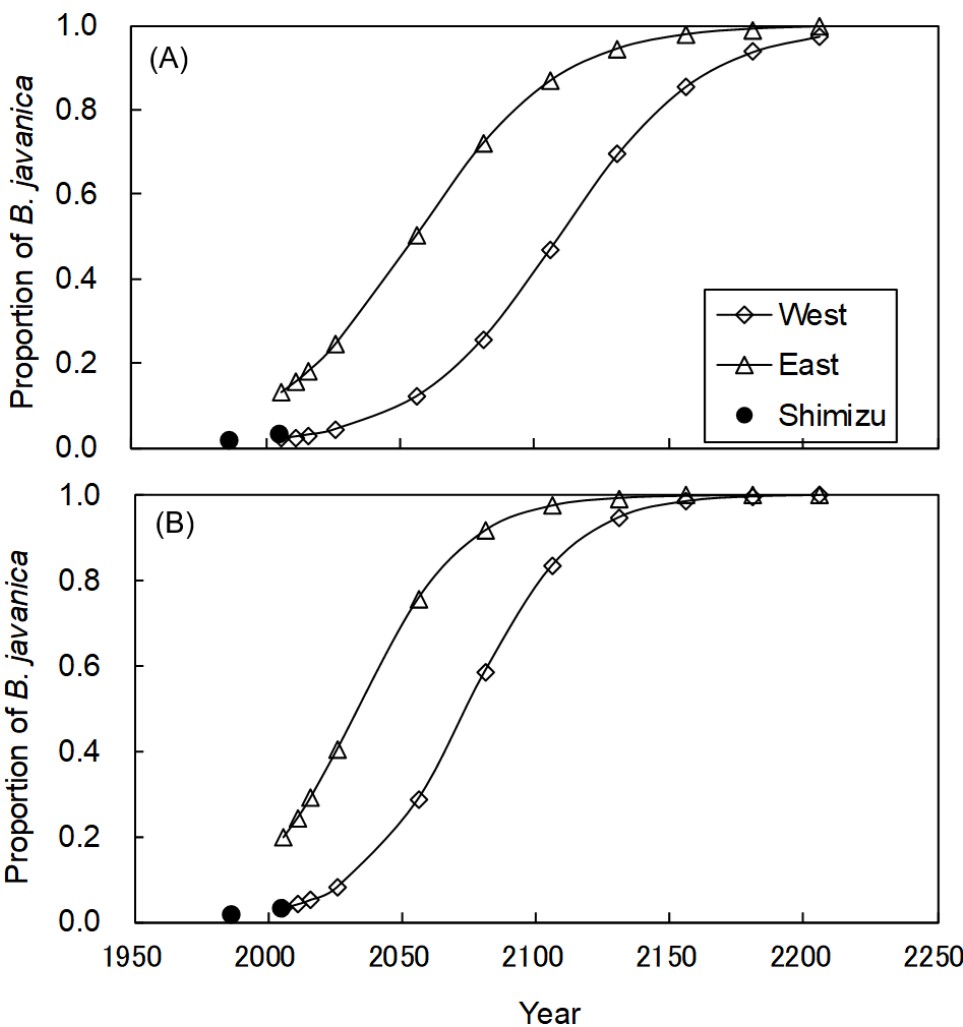

**Figure 8** **Predictions of the increase in *Bischofia javanica* population size.** Estimation of population size is based on (A) the number of stems and (B) the total basal area. Data points were predicted by logistic regressions based on data recorded in 1987 and 2006 in the Shimizu plot (filled circle). "West" and "East" refer to the two plots in Fig. S1.

coexistence (*Kohyama, 1992*), invasion by *B. javanica* that outcompetes all other canopy tree species, such as *E. photiniipholius* and *P. umbellifera*, would homogenize the various crown–understory relationships and disrupt the stable coexistence mechanism of native tree species. *B. javanica* showed positive population growth after the typhoon and a high rate of diameter growth in both canopy and understory individuals compared with those of native species, which would be an important mechanism in the replacement of native forest by an invasive tree species.

Since *B. javanica* has a characteristic of being dominant in the moist forests in Hahajima Island (*Yamashita et al., 2003*; *Tanaka et al., 2010*), it is very likely to expand in the Sekimon. For example, Mt. Kuwanoki in Hahajima Island was the primary mesic forest as Shimon before the war, but after the return from USA, it changed to the forest dominated by *B.*

*javanica* (*Shimizu, 1988*; *Toyoda, 2003*). It is feared that a similar situation will occur at Shimon. The logistic regression curves suggested that *B. javanica* was currently in Phase II (expansion) of its invasion, based on the results of *Webster & Wangen (2009)*, and eradication will be difficult during this phase. The present eradication plan of the Forest Agency prescribes that less than 30% of the total volume can be removed to prevent soil erosion. Our logistic model predicted that *B. javanica* would account for 30% of the basal area by 2017 in the eastern plot and by 2057 in the western plot. These estimations provide important time limits at which it is possible to eradication all mature individuals at once, in compliance with the guideline. In other forests on Haha-jima Island, *B. javanica* has become the dominant tree species (40% to 50% of all individual stems or relative dominance) and has affected plant species diversity (*Shimizu, 1988*; *Toyoda & Kawaoka, 2005*). In addition, this dominance range (30% to 50%) corresponds to the stage of most rapid expansion in population size represented by the logistic curve. Therefore, these dominance values are considered to be useful to set a time limit for action to eradicate both empirically and logically. It is of crucial scientific importance that the population growth rate of invasive tree species can be estimated for a primary forest of high conservation value.

## CONCLUSIONS

This study presents a typical example of the expansion mechanism and quantitative prediction of the time-limit to eradicate an invasive tree species in an insular primary forest. The differences in diameter growth rates among tree species and the relationships with crown position explained the mechanism by which *B. javanica* outcompetes and excludes many of the native tree species. Understory individuals of *B. javanica* grew more rapidly than native tree species and, once reaching the forest canopy, suppressed the growth of native species, resulting in their gradual decline. This pattern of competition also explains how invasive tree species reduce species diversity in natural forests. Prediction by a simple logistic regression model suggested the urgent need for eradication and will contribute to decision-making to develop an effective conservation strategy (*Higgins, Richardson & Cowling, 2000*; *Buckley, Briese & Rees, 2003*). The short settlement history (about 200 years) of the Ogasawara Islands has allowed the primary forests to survive and retain many endemic endangered plants as in the case of the Sekimon forests (*Abe, Tanaka & Shimizu, 2018*). Since the impacts of alien trees appears with a time-lag, however, the impacts confirmed in this study is likely to be even greater (*Downey & Richardson, 2016*). Immediate eradication of *B. javanica* and long-term monitoring are required to prevent further degradation of biodiversity in the Ogasawara Islands.

## ACKNOWLEDGEMENTS

We thank the staff of the National Forest Division of the Ogasawara General Office and Ministry of the Environments for granting permission to carry out our field survey. Yoshio Hoshi and Hiromi Umeno helped with the field surveys. We thank Robert McKenzie, PhD, from Edanz Group, for editing a draft of this manuscript.

### Funding
This study was funded by the Japanese Ministry of the Environment (Global Environmental Research Coordination System). The funders had no role in study design, data collection and analysis, decision to publish, or preparation of the manuscript.

### Grant Disclosures
The following grant information was disclosed by the authors:
Japanese Ministry of the Environment.

### Competing Interests
The authors declare there are no competing interests.

### Author Contributions
- Tetsuto Abe conceived and designed the experiments, performed the experiments, analyzed the data, prepared figures and/or tables, authored or reviewed drafts of the paper, and approved the final draft.
- Nobuyuki Tanaka and Yoshikazu Shimizu conceived and designed the experiments, authored or reviewed drafts of the paper, and approved the final draft.

### Field Study Permissions
The following information was supplied relating to field study approvals (i.e., approving body and any reference numbers):
Field survey was approved for the Ogasawara National Park by the Ministry of the Environment (No.0606328007, No.080507006) and for the Ogasawara National Forest by the Forest Agency (No.18-2-50 and No.20-1-32).

### Data Availability
The plot data is available at figshare: Abe, Tetsuto; Tanaka, Nobuyuki; Shimizu, Yoshikazu (2020): Data.xlsx. figshare. Dataset. https://doi.org/10.6084/m9.figshare.12051501.v1.

### Supplemental Information
Supplemental information for this article can be found online at http://dx.doi.org/10.7717/peerj.9573#supplemental-information.

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
