# Peer review of "Outstanding performance of an invasive alien tree Bischofia javanica relative to native tree species and implications for management of insular primary forests"

_PeerJ, doi:10.7717/peerj.9573_

## Round 0.1 · original submission · Major Revisions

I have received the reports from three reviewers. They think this work is interesting but also raise many comments and suggestions. Please carry out careful revision based on these comments and suggestions.

Reviewer 1 ·

Basic reporting

This study aims to understand the mechanisms of Bischofia javanica invade into the primary forest in Haha-jima Island, Japan. The English used in the manuscript is easy to understand, and introduction of the study is also clear. One concern is the definition of invasive species. A study in a nearby island, Taiwan, showed a primary forest on limestone was dominated by both Pisonia umbellifera and Bischofia javanica (Lin et al., 2017). Therefore, can the growing pattern of the Bischofia javanica considered as a northward expansion of the territory of the species, rather than invasive? If this expansion is a cause of natural climate change, such as global warming, is it really necessary to eradicate the species? How can we justify invasion when considering the regional flora, as Bischofia javanica is actually widely distributed in the east and southeast Asia, including Japan. Therefore, it is vital to define the definition of invasive at the introduction of the manuscript but also acknowledge that the species is actually widely distributed in the neighbourhood region.

The crown width index is not clearly defined. I think it is necessary to explain the rationale of the authors to use the index in the method section. For example, what the ecological meaning of the index is, and what a high number of the index indicates. Is the index necessary to indicate crown width, can it imply something else, such as coexistence potential? As the ecological meaning is not clear, it may be easier for the readers if the authors use a straight forward name of the index (e.g., covered stem per canopy tree?). However, again, what is the ecological meaning if more stems were covered under the crown of a species? Does this really correlate with crown width?

In Fig.4, 5, and 6, similar to the previous comment, the use of ‘above’ and ‘below’ is not clear. Please use a straight forward term, e.g., ‘covered by’ ‘be covered’. Also, in Fig 5 and 6, please clearly indicate which ones are the canopy species, and which ones are the understorey species in the figure legends.

Experimental design

The growth of large trees for one cm would have a different meaning in the total quantity of biomass compared to that of small trees. The authors need to justify why did they choose to use absolute growth rate, rather than the relative growth rate in the method section.

In Fig. 2, it seems to be the changes in density. As there are two kinds of growth, one is density changes and the other is diameter growth, it is better to clarify the terms used in the study.

Line 154 did the authors studied seedlings? In the method section, they only mentioned a survey of trees with DBH ≥ 10 cm in 2006 in line 115. What is the definition of seedlings in this study?

Line 179 to 183 how did the authors calculate the mortality rate? Please clarify in the method. There are several calculating methods, please see Kohyama et al. (2017).

Validity of the findings

In Fig.8 it is not clear why there are only two points of census data, but can have a prediction for two separate plots. If one point indicates the 1987 census and the other indicates the 2006 census, there should be four points in order to predict the patterns for two separate plots. Please explain.

Additional comments

Line 125 ‘……a map of individual trees created……’ revise the sentence as ‘ a tree-by-tree map drawn’

Line 266 the ecological meaning of this index is not clear.

Kohyama, T.S., Kohyama, T.I., Sheil, D., 2017. Definition and estimation of vital rates from repeated censuses: Choices, comparisons and bias corrections focusing on trees. Methods in Ecology and Evolution. 9, 809–821.

Lin, Y.-C., Comita, L.S., Johnson, D.J., Chen, M.R., Wu, S.H., 2017. Biotic vs. abiotic drivers of seedling persistence in a tropical karst forest. J. Veg. Sci. 28, 206-217.

·

Basic reporting

This is a very good paper that provides robust data, analysis and interpretation to show how an alien tree species is able to take advantage of a major disturbance to assume dominance of a biodiverse forest with important implications for biodiversity conservation.

I found the paper to be well conceived, well written, well referenced. In all, a very useful contribution to the growing literature on biological invasions in forests and alien tree invasions.

Experimental design

The study took advantage of a natural event and thus uses a natural experiment - obviously it would have been better to have replicates etc, but that was not possible. Overall, I think that the way that data were collected, analysed and interpreted is both novel and adequate to arrive at the conclusions that they did.

Validity of the findings

Although several studies have reported on the major impacts of invasive trees on islands (these are adequately referenced in the paper), this one makes especially interesting (and very worrying) new observations.

Additional comments

Minor specific suggestions:

line 34 - suggest replacing "invasive tree" with "tree invasions" (the latter term is more widely used)
line 46 - this species is currently known as Morella faya
lines 300-301 - The authors may want to consult the framework of Downy & Richardson (2016) [Downey, P.O. & Richardson, D.M. (2016). Alien plant invasions and native plant extinctions: a six-threshold framework AoB Plants 8: plw047; doi: 10.1093/aobpla/plw047] when discussing how the impacts reported in this study could results in more dramatic impacts in terms of biodiversity loss over time.
line 355 - author name = Higgins
Figures 2-8 - Link to abbreviated names - I suggest that the full names and abbreviated names are given in Table 1- readers should not need to consult online appendices to get the full names. In all figures I suggest that the focal species (Bischofia javanica) be indicated in a different shading to make it stand out.
Table 1 - the terms "indigenous" and "alien" need to be explained here - indigenous/alien to which (bio)geographical region?. Bischofia javanica should be highlighted to make it stand out clearly.

Reviewer 3 ·

Basic reporting

no comment

Experimental design

no comment

Validity of the findings

no comment

Additional comments

In the manuscript, authors compared population dynamics (stem growth, survival and population changes) between an invasive tree, B. javanica and native tree species in primary native forests in an oceanic island to evaluate processes of its invasion and expansion. In addition, authors constructed logistic regression model using long-term census data to predict future expansion of B. javanica. Clarifying invasion processes of invasive trees to native forests and predict future invasion is very important to conduct effective conservation actions, especially, oceanic island ecosystems. Therefore, it is my opinion that this article warrants publication with major revision. There are several criticisms in this manuscript, but these could be remedied.

General comments
In this study, authors suggested that shading by canopy of B. javanica can inhibit growth of understory trees of native species. However, authors did not measure light environments directly. Therefore, discussions about effects of canopy of B. javanica on understory trees would be speculation. Authors should revise them.

I can not judge whether prediction of expansion based on the logistic model is reliable or not. This model is based only on data in 1987 and 2006. Are there direct or indirect evidences that changes in stem density and BA in this period is common in the forests and will continue in the future?

Overall, descriptions of hypotheses, predictions, measurements and statistical analyses are insufficient. I think that the four components and their correspondence should be descripted clearly. For example, correlations between annual diameter growth and population in Results (L194-198) is not descripted in Statistical analyses.

Specific comments
Line 28
“inhibit” is inappropriate expression because direct investigation about inhibition of B. javanica to native trees are not conducted.

L38-41, L45-47
These sentences are not necessary because it is likely that effects of B. javanica in this study are direct competitive effects for light, which is not ecosystem engineering.

L140-144
What is explanatory and responsible variables?

L177-183
Statistical analyses about mortality or survival among tree species (at least between B. javanica and dominant species) are necessary. For example, it can be analyzed using by GLM that responsible variable is dead or alive of individual stems (binomial error) and explanatory variable is tree species.

L184-192
Statistical analyses about stem density among tree species (at least between B. javanica and dominant species) are necessary. For example, it can be analyzed using by GLM that responsible variable is numbers of stem (Poisson or negative binomial error) and explanatory variable is tree species.

L194-198
The information is insufficient to show that whether values of B. javanica is deviated from those of other species These results should be also shown by scatter diagram.

L206-208
“inhibit” is inappropriate expression because direct investigation about inhibition of B. javanica to native trees are not conducted.

L232-234
An increase of population size depends not only on growth and survival but also on recruitment. Discussion about recruitment is also necessary.

L234-235:なぜ? 今回の台風に対する反応はあくまで相対的な値では
In this study, discussions about responses to typhoon damages are based on comparisons between B. javanica and native species. This can not suggest that native species do not adapt to typhoon damage. Are absolute growth rates of native species in this study less than those in other regions?

L250-251
Are there any evidences that B. javanica is still in an early stage of invasion?

---

## Round 0.2 · accepted · Accept

Thank you for your revised manuscript and response to the referees. All concerns have fully been addressed. I am delighted to accept your revised manuscript.

·

Basic reporting

I have studied the response of the authors to my previous review and am satisfied that all my concerns were adequately dealt with.

I also read the full paper and believe that it is improved overall and that it makes an important contribution to the literature on tree invasions.

Experimental design

Sound and appropriate

Validity of the findings

Good

Additional comments

I am happy with the responses to my comments and suggestions.